# The Effects of Novel Pulsed Electromagnetic Field Therapy Device on Acute Distal Radius Fractures: A Prospective, Double-Blind, Sham-Controlled, Randomized Pilot Study

**DOI:** 10.3390/jcm12051866

**Published:** 2023-02-27

**Authors:** Shai Factor, Ido Druckmann, Franck Atlan, Yishai Rosenblatt, Daniel Tordjman, Raphael Krespi, Efi Kazum, Tamir Pritsch, Gilad Eisenberg

**Affiliations:** 1Division of Orthopedic Surgery, Tel Aviv Medical Center, Sackler Faculty of Medicine, Tel Aviv University, Tel Aviv 6423906, Israel; 2Division of Radiology, Tel Aviv Medical Center, Sackler Faculty of Medicine, Tel Aviv University, Tel Aviv 6423906, Israel

**Keywords:** distal radius, fracture, union, pulsed electromagnetic field, bone growth stimulation, electrical stimulation therapy

## Abstract

Background: this pilot study aimed at determining whether the application of a novel new method of generating pulsed electromagnetic field (PEMF), the Fracture Healing Patch (FHP), accelerates the healing of acute distal radius fractures (DRF) when compared to a sham treatment. Methods: 41 patients with DRFs treated with cast immobilization were included. Patients were allocated to a PEMF group (*n* = 20) or a control (sham) group (*n* = 21). All patients were assessed with regard to functional and radiological outcomes (X-rays and CT scans) at 2, 4, 6 and 12 weeks. Results: fractures treated with active PEMF demonstrated significantly higher extent of union at 4 weeks as assessed by CT (76% vs. 58%, *p* = 0.02). SF12 mean physical score was significantly higher in PEMF treated group (47 vs. 36, *p* = 0.005). Time to cast removal was significantly shorter in PEMF treated patients, 33 ± 5.9 days in PEMF vs. 39.8 ± 7.4 days in sham group (*p* = 0.002). Conclusion: early addition of PEMF treatment may accelerate bone healing which could lead to a shorter cast immobilization, thus allowing an earlier return to daily life activities and work. There were no complications related to the PEMF device (FHP).

## 1. Introduction

Distal radial fractures (DRF) are among the most common fractures encountered in health care [1]. Annualized estimates in the United States alone suggest an incidence of approximately 640,000 cases, and rising, per year [2]. Closed treatment is the most common method of management, but unstable fractures tend to displace without surgical stabilization [3,4,5]. Recent studies have demonstrated no difference in outcomes, at 12 months between surgical and non-surgical treatment of DRF in the elderly [6,7].

Pulsed electromagnetic field (PEMF) is a modality often used for bone growth stimulation throughout various clinical settings including orthopedic surgery, such as treatment of fracture non-unions [8,9]. Data from many in vitro and in vivo studies demonstrated that PEMF positively effects bone healing by altering voltage-gated ion channels, increasing cytosolic calcium, enhancing early angiogenesis, and promoting osteoblast differentiation and maturation [10]. In addition, one study has demonstrated that PEMF exposure increased cell proliferation, adhesion, and the osteogenic commitment of Mesenchymal stem cells (MSCs), even in inflammatory conditions [11]. The results of the above mentioned studied demonstrate the potential to shorten the healing time for fractures and allow patients to return to normal activities earlier, which can result in shorter recovery time and can be cost-effective for both the patient and the health-care system [12].

The Fracture Healing Patch (FHP) (Pulsar Medtech Ltd., Bnei Brak, Israel) is an external thin and flexible silicone patch that incorporates a power source and micro-electronic modules, which generates a PEMF to enhance fracture healing. The FHP is placed topically under the cast at the fracture site and produces a continues, focused PEMF that affects a fracture region only. The device is disposable, does not require battery charging and works continuously for the duration of the treatment. The FHP device incorporates a battery, coil and electronic modules, all incapsulated in a silicone body, designed as a flexible patch.

The primary aim of the study was to determine whether the application of the FHP generated PEMF as an adjuvant to immobilization for acute DRF, treated non-operatively, will accelerate bone healing. It was hypothesized that PEMF would accelerate the extent of the fracture union by up to 30% as assessed by CT scans. The secondary aim included the effect of PEMF in acute DRFs on functional outcome. Finally, the third aim was reduction in the incidence, severity, and frequency of all Adverse Events (AE), including pressure-related complications such as Volkmann contracture, compartment syndrome, acute carpal tunnel syndrome pressure necrosis of the skin, and/or complex regional pain syndrome.

## 2. Materials and Methods

### 2.1. Study Design

This prospective, double blind, randomized, sham-controlled study was conducted at a level I trauma center between May 2020 and March 2022. Institutional review board approval was obtained for all aspects of this study in accordance with the institutional policies and written informed consent for participation was obtained from every patient.

Inclusion criteria were a closed unilateral dorsally angulated DRF (Colles’) visible by X-ray; indication for non-operative treatment by means of cast immobilization with or without closed reduction; age > 18 years and patients able to adhere to the visit schedule and protocol requirements and be available to complete the study. Patients were excluded if they had intra articular fracture or extra-articular fracture that meets the criteria for operative fracture fixation, presence of hardware in the forearm or hand, previous fractures or bone surgery in the currently fractured side, synovial pseudarthrosis, multiple trauma (several fractures at once), joint diseases that affect the function of the wrist and/or hand of the injured arm, pregnancy or women who are breast-feeding and the presence of a life supporting implanted electronical device.

Eligible patients were randomly assigned to one of the two groups; Group 1 (Active): standard treatment + active FHP and Group 2 (Control): standard treatment + sham FHP. Half of the PEMF devices were not activated at random before the application to the patients. Two types of activators were used: active and sham. Sham activated devices gave outward signs of normal function but did not generate a signal. Treatment allocation was by block randomization, with a block size of four. The randomization was performed after the patients were admitted to the emergency room. Only at the end of the data processing, the serial number of the FHP indicated whether it was an active device or not. The study duration was 12 weeks.

### 2.2. FHP Device

The FHP model used in this trial is comprised of 2 units which are placed on the contralateral sides of the arm (volar and dorsal) (Figure 1). The units communicate with each other and are able to adjust the intensity of the PEMF to conform to different arm dimensions, thus creating a uniform PEMF through the arm. The PEMF generated by the FHP is characterized by a pulse frequency of 20 KHz, cycle frequency of 10 Hz and pulse intensity at fracture site of between 0.05 mT and 0.5 mT.

Both patients and evaluators were blind to whether the FHP device was active or not.

The FHP device was placed under the cast in the ED following reduction if performed and prior to cast application. The FHP was active (group 1) for 24 h a day continuously throughout the study period. At study completion, device serial numbers were used to determine which patients received an active device.

Primary objectives of the study were to determine whether the use of FHP by means of PEMF in acute DRFs will accelerate bone healing.

Secondary objectives included the effect of PEMF in acute DRFs on functional outcome.

Safety objectives included: reduction in the incidence, severity, and frequency of all Adverse Events (AE), including pressure-related complications such as Volkmann contracture, compartment syndrome, acute carpal tunnel syndrome pressure necrosis of the skin, and/or complex regional pain syndrome.

### 2.3. Outcome Measures

Primary outcome was fracture union at 4 weeks based on CT scans. Evaluation of subjective and objective parameters such as pain, function, range of motion (ROM) as well as radiological outcomes was performed at 2, 4, 6 and 12 weeks after the placement of the FHP device. The presence of complications was also noted. All examined parameters were assessed by one of the authors blinded to group allocation.

### 2.4. Radiologic Assessment

All radiographs and computed-tomography (CT) scans were reviewed independently by a musculoskeletal fellowship-trained radiology attendant and two senior orthopedic surgeons who were blinded to study groups. Radiographic healing was defined as the interval in days between the occurrence of the fracture and the time when bridging in three of four cortices is seen on X-ray images. A determination was made at each follow up evaluation by using Radius Union Scoring System (RUSS) score [13].

At 4 weeks, all patients underwent a computed-tomography (CT) scan. All wrist scans were done in the prone position with the fractured hand extended over the head (“superman position”). If the patient was uncomfortable in this position the scan was performed in the supine position with the hand to the side of the patient. All scans were performed on a Brilliance 64-slice MDCT scanner (Philips, Cleveland, OH, USA) using 64.0 × 0.625 mm collimation, and a slice thickness of 1 mm. All scans were non-contrast. Direct multiplanar reformation function was used to generate coronal and sagittal reformations with a slice thickness of 3 mm. All CT scans were interpreted at Picture Archiving and Communications System workstations (Centricity; GE Healthcare, Chicago, IL, USA). The evaluation of the extent of fracture union was performed using CT scans in all three planes: sagittal, coronal, and axial. However, to calculate the percentage of bony bridges, we focused on the axial cuts, which provided a circumferential view of all cortices. The average extent of union was then calculated based on the evaluation of the axial cuts, following the method described by Singh et al. [14,15]. Fractures were categorized as the following: no union (0% to 24% of the continuity of the trabecular bridging across the whole width of the distal radius), partial union (25% to 74% trabecular bridging) or union (75% to 100% trabecular bridging) [15].

### 2.5. Functional Outcomes and Quality of Life Assessment

Pain and function were assessed by the SF-12 [16] survey and patient-rated wrist evaluation (PRWE) [17], before applying the FHP device, at 4, 6 and 12 weeks. The SF12 questionnaire is a valid and reliable instrument to measure pain and psychosocial well-being. The PRWE is a 15-item questionnaire designed to measure wrist pain and disability in activities of daily living. The PRWE allows patients to rate their levels of wrist pain and disability.

### 2.6. Functional Assessments

Pain-free grip: assessment of grip strength via a JAMAR dynamometer [18]. The dynamometer measures in increments of 0.1 kg. The mean of the three measurements, 2 min apart, was considered as the grip strength for a patient at a specific visit. Flexion, extension, radial and ulnar deviation, pronation, and supination range of motion (ROM) were also measured. All tests were compared with the opposite unaffected side.

### 2.7. Safety Outcomes and Rehabilitation

Patients were examined for cast pressure-related complication signs and peripheral oxygen saturation was measured at the injured hand by pulse oximetry. Patients were also evaluated in each clinical visit for signs of: Volkmann contracture, compartment syndrome, acute carpal tunnel syndrome, pressure necrosis of the skin, and/or complex regional pain syndrome. In cases where it was necessary to replace the cast, the same FHP device was kept under the new cast. In all patients, the cast was removed at a maximum of 6 weeks following the injury, and the decision was based on CT scan evaluation. PEMF treatment was discontinued at the day of cast removal. All patients began rehabilitation after cast removal, which consisted of active and active assisted ROM of the wrist and fingers and avoidance of exertion and heavy weightlifting with the injured hand for 6 weeks. At 12 weeks, patients were allowed to start passive activation.

### 2.8. Statistical Analysis

Power analysis was conducted with an expected outcome difference of 30% in the extent of the fracture union assessed by CT at 4 weeks as compared to the control group. The alpha error level was set at 5% (two-sided significance level); power was set at 80%. Including an anticipated dropout rate of 10%, this resulted in a sample size of 23 patients per group. Data were analyzed with IBM SPSS statistics software version 28.0. (SPSS Inc. Headquarters, 233 S. Wacker Drive, 11th floor Chicago, IL 60606, USA). The significance levels were set at 0.05. Baseline characteristics are presented as means and standard errors for continuous variables and as frequencies and percentages for categorical variables. Chi-square tests and independent *t*-tests were performed to compare the two groups for categorical and continuous variables, respectively.

Agreements between raters were tested by the Friedman test. To reduce the within variability in RUSS scale, we choose the mean and the median value from the three raters.

Differences in RUSS scale between the two groups were tested by independent *t*-test.

Differences in the CT results between the two groups were tested by the independent *t*-test.

## 3. Results

A total of 61 patients were screened. Fifty-one (51) patients met the inclusion criteria and were randomized: in nine patients, fracture displacement occurred a week after the treatment initiation, and they underwent surgical treatment. One patient had to discontinue his participation in the study due the other medical condition. The remaining forty-one patients (41 fractures) (12 males, 29 females; mean age 59 years (range 21–88)) made up the core group that adhered to the study protocol and were the basis for inferences regarding the efficacy of the FHP PEMF device. Forty-one fractures were randomly treated with either active FHP or sham FHP device. Three patients and two patients were lost to follow-up in the active and control group, respectively (Figure 2, Table 1).

There was no significant difference between the two treatment groups with regard to any of the patient or fracture-related parameters; therefore, the randomization process produced similar treatment groups for the efficacy comparisons.

### 3.1. Radiological Assessment

Fractures treated with active PEMF demonstrated significantly higher extent of union at 4 weeks as assessed by CT (76% vs. 58%, *p* = 0.02) (Figure 3, Table 2). All raters gave a significantly higher healing percentage to the PEMF treated group, however there was no statistical agreement between orthopedic surgeons and radiologist. Agreement was found between two orthopedic surgeons.

X-rays were evaluated using RUSS by the same blinded reviewers. No statistically significant differences between the groups were found (Figure 4). Additionally, there was no agreement between the reviewers.

### 3.2. Functional Assessment

Time to cast removal was significantly shorter in PEMF treated patients, 33 ± 5.9 days in PEMF vs. 39.8 ± 7.4 days in sham group (*p* = 0.002).

Hand grip strength was measured after a cast removal. At 6 weeks, the mean grip strength in the active group were 7.49 ± 1.84 Kg vs. 6.33 ± 1.86 Kg in the control group (*p* = 0.684). At 12 weeks, the mean grip strength in the active group were 14.22 ± 2.67 Kg vs. 8.25 ± 2.19 Kg in the control group (*p* = 0.114).

### 3.3. Range of Motion

At 12 and 24 weeks, wrist flexion was significantly better in the PEMF treated patients as compared to control group (65° vs. 33°, *p* = 0.012: 64° vs. 20°, *p* = 0.015, respectively). All other parameters were slightly better in PEMF treated group, however not statistically significant (Figure 5).

### 3.4. PRWE

Total PRWE score was better in PEMF treated patients at 12 weeks, however not statistically significant (*p* = 0.07) (Figure 6C). Pain sub-score was better during examination in the PEMF group at week 12 in comparison to that in the control group (14.4 vs. 21.7, *p* = 0.06).

### 3.5. SF 12

SF12 physical score was significantly higher in PEMF treated group at 12 weeks (47 vs. 36, *p* = 0.005) (Figure 7). No differences were noticed in the mental score.

No adverse events or complications attributable to the device, and no contraindications to use of the device were reported during the study. No mechanical or technical difficulties with use of the device were reported by the patients.

## 4. Discussion

The main findings of this study demonstrated that fractures treated with active PEMF demonstrated significantly higher extent of union at 4 weeks as assessed by CT in comparison to control group. Time to cast removal was significantly shorter in PEMF treated patients. Additionally, functional outcomes in terms of SF12 physical score and PRWE score were better in PEMF treated group.

The literature is lacking high methodological quality studies, which investigate the effects of PEMF on acute fracture healing. In this study, DRF was chosen as the model to test the effects of PEMF, since it includes both trabecular and cortical bone, is accessible for radiographs, has little soft tissue that can distort the radiograph, and is amenable to multiple functional and radiological endpoints.

PEMF is reported to be an effective and FDA approved for the treatment of nonunion long-bone fractures. The use of PEMFs in the management of nonunion is indicated only in presence of a valid mechanical environment (appropriate fracture alignment, the limb immobilization, and the lack of a significant bone loss). The success of PEMF therapy in the treatment of non-unions ranges between 73% and 85%, based on fracture and patients’ related factors and patients’ compliance [19]. A study by Murray at al., reported that the time required for fracture healing can be significantly affected by device usage and/or patient compliance [20]. Daily effective dose of PEMF therapy, depends directly on the patient’s adherence to the device. FHP device, used in the current study, is non-invasive, disposable, fully automated, does not require charging and functioning continuously for the entire treatment duration. Its placement under the cast makes it unnoticeable by the patient, thus allowing optimal adherence to the prescribed treatment. Furthermore, there were no adverse events associated with its use.

Currently, PEMF therapy plays a pivotal role in the biophysical stimulation of fracture healing, both alone and as an adjunct to the surgical treatment. The evidence regarding the effects of PEMF in fresh fractures healing is increasing but still limited. The efficacy of PEMF in stimulating bone healing in patients undergoing tibial and femoral osteotomies was demonstrated in two previously published studies [21,22].

There are a few studies looking at the effect of PEMF, Fontanesi et al. in acute tibial fractures [23], and Faldini at al., in femoral neck fractures [24], reported a significant reduction of time to union and an increase in the percentage of fracture healing in PEMF treated patients as compared to controls. These results are supported with the current study findings.

Recently, several studies assessed the effects of biophysical stimulation modalities including PEMF on fresh distal radius fracture healing. Kristiansen at al. tested the efficacy of a low-intensity pulsed ultrasound medical device for shortening the time to radiographic healing of dorsally angulated DRFs that had been treated with manipulation and a cast [25]. They reported a significantly shorter time to union for the fractures treated with ultrasound compared to placebo (61 ± 3 days compared with 98 ± 5 days; *p* < 0.0001). Similarly, to the current study, they concluded that this specific ultrasound signal accelerates the healing of DRFs and decreases the loss of reduction during fracture-healing. Saebo et al. investigated possible effects of photo-biomodulation therapy (PBMT) in DRF during immobilization with semicircular orthopedic cast [26]. Unlike the current study, they found that PBMT administered during the immobilization period of DRF had no effect on perceived pain and function measured through PRWE. It is important to notice that in order to apply the PBMT, the cast and elastic bandage were temporarily removed to gain access to the skin during irradiation, thus causing some discomfort to the patients and probably compromising the fracture alignment. In the current study, the FHP was placed under the cast at the fracture site and remained there until the cast was removed.

A study by Lazovic et al. assessed whether the use of PEMF during cast immobilization of DRF provides beneficial effects on pain, edema, wrist range of motion and function immediately following cast removal [27]. They reported that DRF patients had better results immediately after cast removal with less edema and greater wrist range of motion. A recently published paper by Krzyżańska et al. suggested that the early addition of PEMF treatment during cast immobilization of DRFs has beneficial effects on the pain, exteroceptive sensation, range of motion, and daily functioning of patients [28]. As was previously discussed, the exposure time is vital for the PEMF therapy effectiveness. Thus, to considerably enhance bone healing, the PEMF device should be used for a minimum of 8 h per day, for at least 45–60 days (depending on the fracture and patient’s features) [29]. In the two aforementioned studies [27,28], the PEMF device was applied for 30 min per day which is shorter time than recommended. This may explain why they did not demonstrate enhanced fracture healing. Since the success of PEMF treatment is strongly associated with the daily PEMF dosage and patient compliance, its placement under the cast allows optimal adherence to the prescribed treatment. This continuous targeted stimulation generated by the FHP device for 24 h a day, resulted improved healing at the early stage.

The PRWE score is a well-accepted tool to assess patients’ functional outcomes after DRF [17,30]. The minimum clinically important difference (MCID) is often used as the new standard for determining effectiveness of a given treatment and describing patient satisfaction in reference to that treatment. A study by Walenkamp MMJ et al. determined the MCID of the PRWE score in patients with DRFs [31]. They recommended using an improvement on the PRWE of more than 11.5 points as the smallest clinically relevant difference when evaluating the effects of treatments in studies of DRFs. In the current study, total PRWE was improved by 17.7 points from week 6 to 12 in PEMF treated patients, while PRWE in control group improved by 6.3 point during the same time interval.

### Limitations

The limitations of this study are related to the patients lost and the relatively low numbers reported on. It is possible with larger numbers, the effect seen may have evened out or functional outcomes could have been different. In addition, the nature of the FHP device may have affected the ability to achieve an ideal reduction and cast fixation. However, as both groups used the same device, this limitation has been eliminated in the current study.

## 5. Conclusions

Focused continues PEMF treatment generated by a novel device, the FHP, was safe to use in distal radius fracture treated with a cast It demonstrated a positive effect on fracture healing and positive short-term effects on functional outcomes. Furthermore, there were no complications related to the FHP device. The results of this pilot study suggest that early addition of PEMF treatment may accelerate bone healing, which could lead to a shorter cast immobilization, thus allowing an earlier return to daily life activities and work. Additionally, high compliance to the treatment is expected since the FHP device does not require any action by the patient. This pilot study provides preliminary data on the potential benefits of this novel device, and larger studies are warranted to validate these findings.

## Figures and Tables

**Figure 1 jcm-12-01866-f001:**
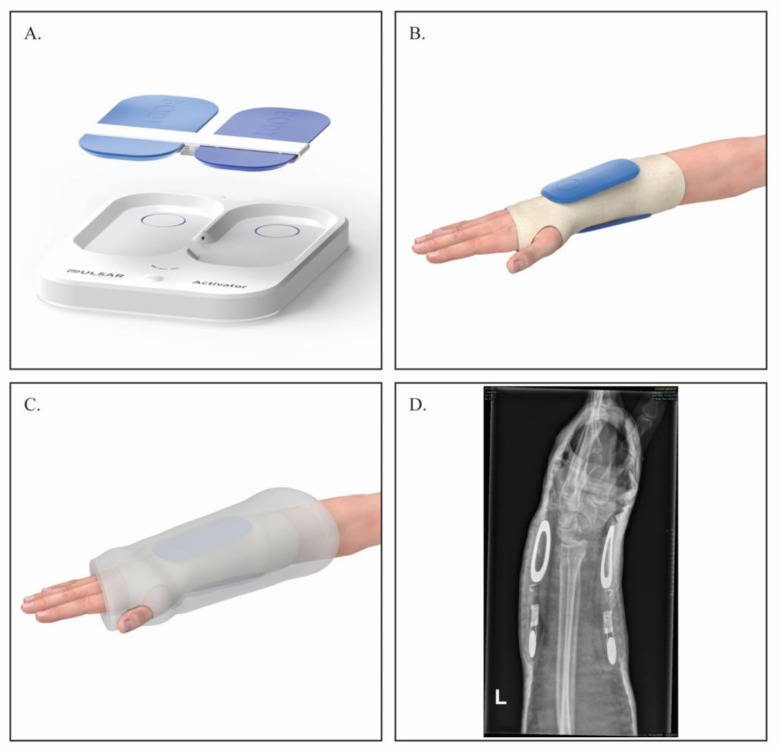
The treatment equipment used in the study. (**A**) A set of Pulsar FHP and activator; (**B**) activated or sham FHP placed on the patient’s hand over the Velband dressing; (**C**) schematic presentation of the FHP under the cast; (**D**) representative X-ray, of the FHP placed under the cast (lateral view).

**Figure 2 jcm-12-01866-f002:**
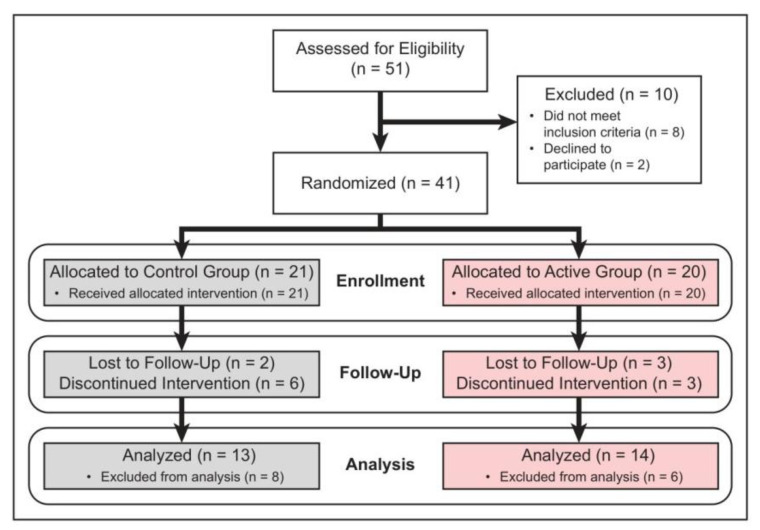
Study consort flow diagram demonstrating the method of patient recruitment.

**Figure 3 jcm-12-01866-f003:**
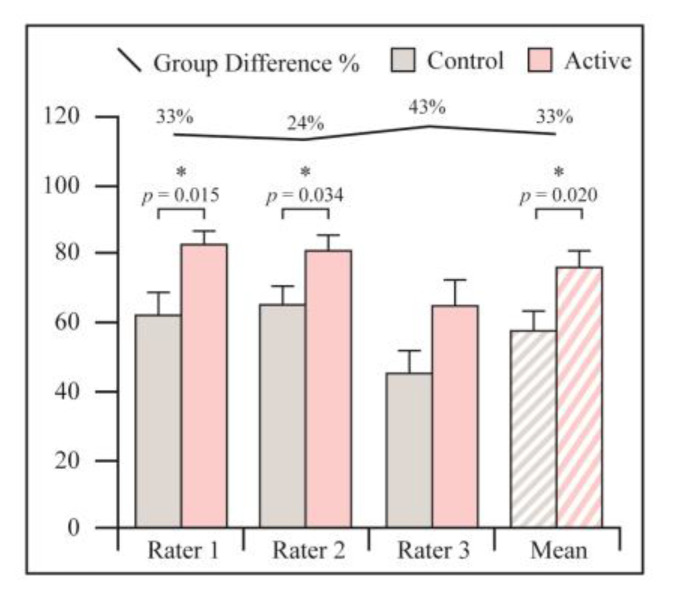
Radiological assessment of percentage of the extent of fracture union at 4 weeks as assessed by CT. Graphs are reported as mean ± SE. Student’s *t*-test. * Statistically significant.

**Figure 4 jcm-12-01866-f004:**
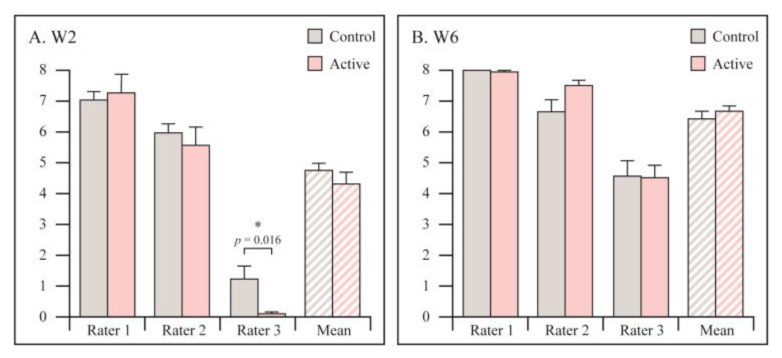
Radiological assessment of fracture healing at 2 and 6 weeks by X-rays, using RUSS. Graphs are reported as mean ± SE. Student’s *t*-test. (**A**) At 2 weeks follow-up. (**B**) At 6 weeks follow-up.* Statistically significant.

**Figure 5 jcm-12-01866-f005:**
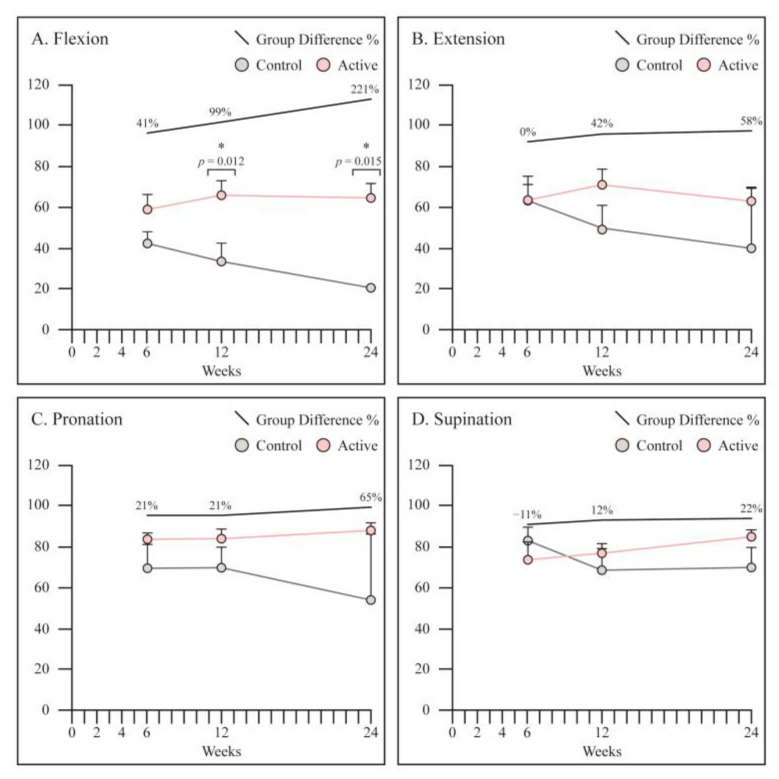
Assessment of functional outcome in terms of wrist range of movement as measured at 6, 12 and 24 weeks. (**A**) Flexion was significantly better in the PEMF treated patients at 12 and 24 weeks. (**B**–**D**) All other parameters were slightly better in PEMF treated group, however not statistically significant. Graphs are reported as mean ± SE. Student’s *t*-test. * Statistically significant.

**Figure 6 jcm-12-01866-f006:**
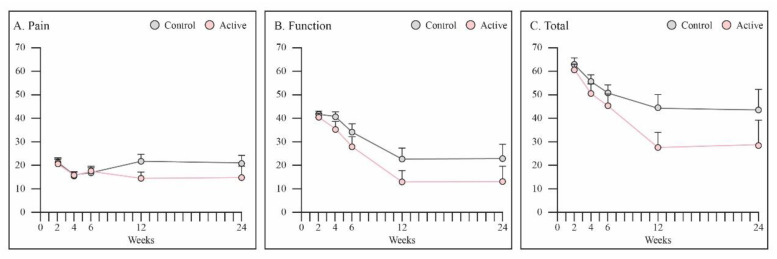
Assessment of functional outcome using PRWE. (**A**) Pain sub-score was better in the PEMF group at week 12 (*p* = 0.06). (**B**) No differences were found in the function sub-score between groups. (**C**) Total PRWE score was better in PEMF treated patients at 12 weeks, however not statistically significant (*p* = 0.07). Graphs are reported as mean ± SE. Student’s *t*-test.

**Figure 7 jcm-12-01866-f007:**
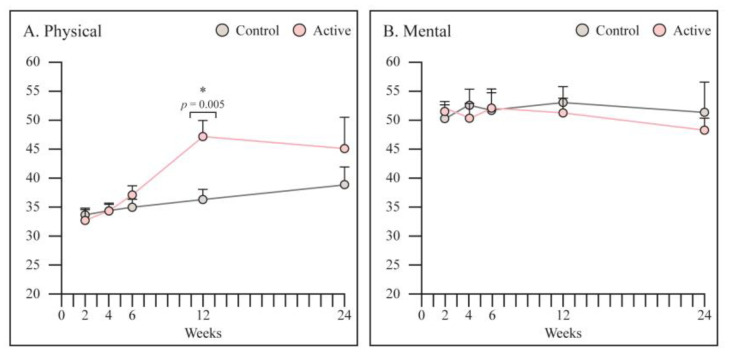
Assessment of patients physical and mental wellbeing using SF 12. (**A**) SF12 physical score was significantly higher in PEMF treated group at 12 weeks. (**B**) No differences were noticed in the SF12 mental score. Graphs are reported as mean ± SE. Student’s *t*-test. * Statistically significant.

**Table 1 jcm-12-01866-t001:** Patients’ demographics.

Variable	Control Group *n* = 21	Active Group *n* = 20
**Age (years)**	59 (range 21–88)	58 (range 26–79)
**Male (*n*, %)**	7 (33)	5 (25)
**Fracture in dominant hand (*n*, %)**	8 (38)	11 (55)
**Fracture type (*n*, %)**		
**Frykman Type (I)**	5 (24)	6 (30)
**Frykman Type (II)**	2 (9)	3 (15)
**Frykman Type (III)**	1 (5)	4 (20)
**Frykman Type (IV)**	3 (14)	1 (5)
**Frykman Type (V)**	0 (0)	1 (5)
**Frykman Type (VI)**	1 (5)	0 (0)
**Frykman Type (VII)**	1 (5)	0 (0)
**Frykman Type (VIII)**	1 (5)	0 (0)
**Smoking (*n*, %)**	4 (19)	3 (15)
**Osteoporosis (*n*, %)**	3 (14)	4 (20)
**Corticosteroids (*n*, %)**	0 (0)	1 (5)

**Table 2 jcm-12-01866-t002:** Fractures categorization by extent of the union at 4 weeks.

Group	No Union (0–24%)	Partial Union (25–74%)	Union (75–100%)
**Control**	1	9	3
**Active**	0	5	9

## Data Availability

The data that support the fundings of this study are available from the corresponding author, upon reasonable request.

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
