# Peer review of "The Effects of Novel Pulsed Electromagnetic Field Therapy Device on Acute Distal Radius Fractures: A Prospective, Double-Blind, Sham-Controlled, Randomized Pilot Study"

_jcm, 2023, doi:10.3390/jcm12051866_

Round 1
Reviewer 1 Report (Previous Reviewer 1)
About the previous comment #2. The author described “The evaluation of the extant of fracture union was performed in each of the axial cuts,". Does this mean that fracture union was not evaluated in the sagittal or coronal direction?
Author Response
Reply to the reviewer.
Original Reviewers’ comments are copied here in italics.
Reviewer comments:
About the previous comment #2. The author described “The evaluation of the extant of fracture union was performed in each of the axial cuts,". Does this mean that fracture union was not evaluated in the sagittal or coronal direction?
Response: We thank the reviewer for this valuable comment. we would like to clarify that although we did evaluate sagittal and coronal images, the axial cuts were used to calculate the exact amount of union. This is because the axial cuts provide us with a circumferential view of all cortices, which helps us to compute the percentage of bony bridges.
We agree that it is important to evaluate fracture union in multiple planes, and we did perform this evaluation. However, the axial cuts provided the most comprehensive view for our analysis. We have updated the manuscript to clarify this point and to better reflect the evaluation of fracture union in all three planes.
Page 8, line 159 and below:
“The evaluation of the extent of fracture union was performed using CT scans in all three planes: sagittal, coronal, and axial. However, to calculate the percentage of bony bridges, we focused on the axial cuts, which provided a circumferential view of all cortices. The average extent of union was then calculated based on the evaluation of the axial cuts, following the method described by Singh et al.”
Thank you again for your valuable feedback, which has helped us to improve the clarity and accuracy of our work.
Reviewer 2 Report (Previous Reviewer 2)
You have revised it several times and packed it into a good article, interesting and novel
Author Response
We wish to thank the reviewer for the in‑depth analysis of our work and for raising several important points that needed clarification. We appreciate the time and effort expended on our behalf.
This manuscript is a resubmission of an earlier submission. The following is a list of the peer review reports and author responses from that submission.
Round 1
Reviewer 1 Report
This study evaluated the fracture healing of distal radius with or without generating pulsed electromagnetic field (PEMF), the Fracture Healing Patch. It was found fratures treated with PEMF demonstrated significantly higher extent of union at 4 weeks. SF12 mean physical score was significantly higher in PEMF treated group. Time to cast removal was significantly shorter in PEMF treated patients. Using this device may be useful in promoting healing of fractures. However, to demonstrate the advantage of using this device, the authors need to address several concerns listed below.
1. If the equipment needs to place under the cast, how to maintain the reduction shape with the cast. Sometime, the cast must be indented to keep the reduction position. Authors need to describe the amount of reduction loss during the course of treatment. Since the authors evaluated X-rays at specific time periods, it would be better to present the results of the X-ray parameters measurements: palmar tilt, radial tilt, and ulnar variance.
2. How did the authors define the fracture union? Although the authors described to use RUSS for the evaluation, it would be better to describe clear definition of fracture union with X-ray evaluations. In addition, the authors described “The evaluation of the extant of fracture union was performed in each of the axial cuts”(line 143-144). Was it possible to evaluate the fracture union of the sagittal direction?
3. Line 226. Time to cast removal was significantly shorter in PEMF treated patients… How did the authors decide to remove the cast? Based on the X-ray union or CT scan? If there is no protocol for removing the cast, the timing of removal may vary from doctor to doctor.
4. To demonstrate superiority over conventional treatment, one additional group with conventional casts may be needed. The authors should justify conducting clinical trials with two groups.
5. It would be better to mention cost-effectiveness of using this device in the discussion.
Author Response
Reviewer 1 comments:
We wish to thank the reviewers for the in‑depth analysis of our work and for raising several important points that needed clarification. We appreciate the time and effort expended on our behalf. We addressed each issue that was raised as follows:
- If the equipment needs to place under the cast, how to maintain the reduction shape with the cast. Sometime, the cast must be indented to keep the reduction position. Authors need to describe the amount of reduction loss during the course of treatment. Since the authors evaluated X-rays at specific time periods, it would be better to present the results of the X-ray parameters measurements: palmar tilt, radial tilt, and ulnar variance.
Response: we thank the reviewer for this valuable and significant comment. The flexibility of the FHP model used in this trial may not be suitable for very complex cases in which high precision reduction is needed. However, in the vast majority of cases, the flexibility of the FHP device enables indenting the cast to keep the reduction position. The quality of the reduction was not compromised because of the device. No reduction was lost during the study due to the use of the FHP device.
Although we did not observe any significant disturbances during the treatment in the emergency department, we addressed this issue in the limitations section of the study.
Page 17, Line 345 and below:
“The limitations of this study are related to the patients lost and the relatively low numbers reported on. It is possible with larger numbers, the effect seen may have evened out or functional outcomes could have been different. The flexibility of the FHP model used in this trial may not be suitable for very complex cases in which high precision reduction is needed. However, in the vast majority of cases, the flexibility of the FHP device enables indenting the cast to keep the reduction position. The quality of the reduction was not compromised because of the device. No reduction was lost during the study due to the use of the FHP device.”
Unfortunately, we do not have data to compare the different angles for each patient and assess the accurate amount of fracture displacement. As indicated in the studies’ methods and results sections, some patients experienced significant worsening of their fracture position, to the point where surgical treatment was recommended and their treatment and participation in the study was discontinued. They were not included in the final analysis. We believe that, for the purpose of the hypothesis we wanted to investigate in this article, the fact that both groups used the same device reduces the chance of bias and allows us to analyze the effect of PEMF on fracture healing as accurately as possible.
- How did the authors define the fracture union? Although the authors described to use RUSS for the evaluation, it would be better to describe clear definition of fracture union with X-ray evaluations. In addition, the authors described “The evaluation of the extant of fracture union was performed in each of the axial cuts” (line 143-144). Was it possible to evaluate the fracture union of the sagittal direction?
Response: we thank the reviewer for this question. Fracture union was defined in the methods section, based both on X-ray evaluation and CT. Primary outcome was fracture union at 4-weeks based on CT scans.
Page 8 line 163 and below:
“The evaluation of the extant of fracture union was performed in each of the axial cuts, then calculating the average [14]. The extent of union was quantified as described by Singh et al [15] . Fractures were categorized as following: no union (0% to 24% of the continuity of the trabecular bridging across the whole width of the distal radius), partial union (25% to 74% trabecular bridging) or union (75% to 100% trabecular bridging) [15].”
- Line 226. Time to cast removal was significantly shorter in PEMF treated patients… How did the authors decide to remove the cast? Based on the X-ray union or CT scan? If there is no protocol for removing the cast, the timing of removal may vary from doctor to doctor.
Response: we would like to thank the reviewer for pointing out this point which needs clarification. Indeed, in common practice, the decision for cast removal is usually physician dependent and based on clinical and radiological parameters, although usually in most cases cast removal is after ~6 weeks following a fracture. In this study, primary outcome was fracture union at 4-weeks based on CT scans, therefore the decision was based on CT scan evaluation.
Page 9, Line 189 and below:
“In cases where it was necessary to replace the cast, the same FHP device was kept under the new cast. In all patients cast was removed at a maximum of 6 weeks following the injury, and the decision was based on CT scan evaluation. PEMF treatment was discontinued at the day of cast removal.”
- To demonstrate superiority over conventional treatment, one additional group with conventional casts may be needed. The authors should justify conducting clinical trials with two groups.
Response: we would like to thank the reviewer for pointing out this point which needs clarification. We addressed this issue in the limitations section. Although, the usual conservative treatment of DRF consists of just cast fixation, without any device or patch beneath it, in this study, in order to eliminate the effect of the FHP, both groups had it under the cast.
Page 17, line 348 and below:
“In addition, the nature of the FHP device may have affected the ability to achieve an ideal reduction and cast fixation. However, as both groups used the same device, this limitation has been eliminated in the current study.”
- It would be better to mention cost-effectiveness of using this device in the discussion.
Response: we thank the reviewer for this suggestion. This study’s’ purpose was to evaluate the safety and efficacy of the FHP. We agree that further studies are needed to better understand and estimate the cost-effectiveness of this device.
Reviewer 2 Report
This study investigated the benefit of pulsed electromagnetic field therapy for fractures of the distal radius
There is indeed an interest in knowing how to enhances healing, and although the distal radius is not a major problem, it could be used as a model.
The title is good, but should include the strength of a sham treatment
Abstract
L19 This is not a cast, but a sham treatment.
Sham treatment should be mentioned in the methods section
Results should state the loss to follow-up
Conclusion should be toned down as the number of patients does not allow clear conclusions to be drawn
In the introduction
What are the problems with shorter recovery time? Socioeconomics ? only one model is used
Some context is missing
What is a bone that heals clinically?
In the methods section
-How did you perform the randomization? When? Who performed it? Where, in the emergency room?
-Could you feel the EM pulse when it was activated?
-Why did you perform a numerical analysis on 20 patients to find a statistical difference?
-Why did you measure grip strength but not report the results?
-What about the analysis of the comparability of the groups after the "lost to follow-up" in 13-14 patients?
What is the main outcome? 4 weeks, 6 weeks? You can not draw conclusions if you have a vague question
In the results section
What is group A or B?
Could you show the analysis between observers.
In the discussion section
The discussion should begin with the most important result of the study
Analyze all the results and explain your hypothesis about the result, why the functional results are different?
There are many more limitations to mention
Conclusion section
It should be softer
Author Response
Reviewer 2 comments:
We wish to thank the reviewers for the in‑depth analysis of our work and for raising several important points that needed clarification. We appreciate the time and effort expended on our behalf. We addressed each issue that was raised as follows:
The title is good but should include the strength of a sham treatment
Response: we thank the reviewer. The title was corrected accordingly and now reads as following:
Focused Continuous Pulsed Electromagnetic Field Therapy Enhances Healing of Acute Distal Radius Fractures: prospective, double blind, sham controlled, randomized trial.
Abstract
L19 This is not a cast, but a sham treatment. Sham treatment should be mentioned in the methods section. Results should state the loss to follow-up. Conclusion should be toned down as the number of patients does not allow clear conclusions to be drawn.
Response: we thank the reviewer for the comment. Abstract was edited accordingly. The data regarding the loss of follow-up was added and a reference to Figure 1 was added in the results section. (Line 221)
Now the abstract reads as following:
Page 3, Line 30 and below:
Background: This study aimed at determining whether the application of a novel new method of generating pulsed electromagnetic field (PEMF), the Fracture Healing Patch (FHP), accelerates healing of acute distal radius fractures (DRF) both clinically and radiologically when compared to a sham treatment.
Methods: 41 patients with DRFs treated with cast immobilization were included. Patients were allocated to a PEMF group (n =20) or a control (sham) group (n =21). All patients were assessed with regard to functional and radiological outcomes (X-rays and CT scans) at 2,4, 6 and 12 weeks.
Results: Fractures treated with active PEMF demonstrated significantly higher extent of union at 4 weeks as assessed by CT (76% Vs 58%, p=.02). SF12 mean physical score was significantly higher in PEMF treated group (47 VS 36, p=.005). Time to cast removal was significantly shorter in PEMF treated patients, 33 ±5.9 days in PEMF VS 39.8 ± 7.4 days in Sham group(p=.002).
Conclusion: Early addition of PEMF treatment may accelerate bone healing which could lead to a shorter cast immobilization, thus allowing an earlier return to daily life activities and work. There were no complications related to the PEMF device (FHP).
In the introduction
What are the problems with shorter recovery time? Socioeconomics? only one model is used
Some context is missing.
Response: we thank the reviewer for pointing this important issue. Comment accepted and the introduction was revised accordingly.
Page 4, Line 66 and below:
“The results of the above mentioned studied demonstrate the potential to shorten the healing time for fractures and allow patients to return to normal activities earlier, which can result in shorter recovery time and can be cost-effective for both the patient and the health-care system”
What is a bone that heals clinically?
Response: we thank the reviewer for this question. Several studies are cited in the text, one main study is a systematic review and meta‐analysis of randomized controlled trials, by Peng et al. they reported that PEMF treatment would increase overall healing rate with risk ratio of 1.22 (95% CI = 1.10–1.35). subgroup analysis was also conducted, showing increased healing rate in both long (femur, tibia, radius) and irregular bones. Additionally, in terms of pain, the pooled results of seven studies indicated that PEMF can alleviate pain.
Peng L, Fu C, Xiong F, Zhang Q, Liang Z, Chen L, He C, Wei Q. Effectiveness of Pulsed Electromagnetic Fields on Bone Healing: A Systematic Review and Meta-Analysis of Randomized Controlled Trials. Bioelectromagnetics. 2020 Jul;41(5):323-337. doi: 10.1002/bem.22271. Epub 2020 Jun 3. PMID: 32495506.
In the methods section
-How did you perform the randomization? When? Who performed it? Where, in the emergency room?
Response: we thank the reviewer for this valuable comment. We added this information to the text.
Page 6, Line 113 and below:
“The randomization was performed after the patients were admitted to the emergency room. Only at the end of the data processing, the serial number of the FHP indicated whether it was an active device or not.”
Page 6, Line 144 and below:
All examined parameters were assessed by one of the authors blinded to group allocation.
Page 6, Line 147 and below:
All radiographs and computed-tomography (CT) scans were reviewed independently by a musculoskeletal fellowship-trained radiology attending and two senior orthopedic surgeons who were blinded to study groups.
-Could you feel the EM pulse when it was activated?
Response: we thank the reviewer for this question.
Patients do not feel the pulsed magnetic field during PEMF therapy. The intensity of the magnetic field is very low and cannot to be felt by the patient.
-Why did you perform a numerical analysis on 20 patients to find a statistical difference?
Response: we thank the reviewer for this comment. Indeed, our study consists of a small group of patients, which we addressed in the Limitations section. Statistical analysis was performed by a statistician, and even with the small sample size, statistically significant differences were found.
-Why did you measure grip strength but not report the results?
Response: we thank the reviewer for this comment. The results regarding grip strength are in the results section, under the subheading “Functional assessment.”
Page 12, Line 245 and below:
“Hand grip strength was measured after a cast removal. No statistically significant differences were found between the groups.”
-What about the analysis of the comparability of the groups after the "lost to follow-up" in 13-14 patients?
Response: we thank the reviewer for pointing this important issue. The primary outcome was fracture union at 4-weeks based on CT scans. The data and statistical analysis were performed on patients in both group who completed the study protocol and were available for the final follow up.
What is the main outcome? 4 weeks, 6 weeks? You cannot draw conclusions if you have a vague question.
Response: we thank the reviewer for this question. Comment accepted and addressed.
The primary outcome was defined, and text now reads:
Page 8, line 141 and below:
“Primary outcome was fracture union at 4-weeks based on CT scans.”
In the results section
What is group A or B?
Response: we thank the reviewer for this question. The information is detailed in the methods section. Page 6, Line 108:
“Eligible patients were randomly assigned to one of the two groups; Group 1 (Active): Standard treatment + Active FHP and Group 2 (Control): Standard treatment + Sham FHP.”
We have edited Table 2.
|
Group |
No union (0%-24%) |
Partial union (25%-74%) |
Union (75%-100%) |
|
Active |
0 |
5 |
9 |
|
Control |
1 |
8 |
3 |
Could you show the analysis between observers.
Response: we thank the reviewer for this valuable comment. Agreements between raters were tested by the paired t-test. To reduce the within variability in RUSS scale, we choose the mean and the median value from the 3 raters. Agreement between the two orthopedic surgeons was statistically significant, however, agreement between them and the radiologist was low. This data is shown in Figure 3.
figure see the attachment.
In the discussion section
The discussion should begin with the most important result of the study
Analyze all the results and explain your hypothesis about the result, why the functional results are different?
Response: we thank the reviewer for the comment. Discussion was edited accordingly.
Page 14 line 267 and below:
“The main findings of this study demonstrated that fractures treated with active PEMF demonstrated significantly higher extent of union at 4 weeks as assessed by CT in comparison to control group. Time to cast removal was significantly shorter in PEMF treated patients. Additionally, functional outcomes in terms of SF12 physical score and PRWE score were better in PEMF treated group.
Page 16 line 339 and below:
“...They recommended using an improvement on the PRWE of more than 11.5 points as the smallest clinically relevant difference when evaluating the effects of treatments in studies of DRFs. In the current study, total PRWE was improved by 17.7 points from week 6 to 12 in PEMF treated patients, while PRWE in control group improved by 6.3 point during the same time interval.
This functional difference may be explained by the fact that PEMF-treated patients removed their casts earlier and started their rehabilitation earlier as well “
There are many more limitations to mention.
Response: we thank the reviewer for the comment. Limitation section was edited accordingly.
Page 17 line 346 and below:
“The limitations of this study are related to the patients lost and the relatively low numbers reported on. It is possible with larger numbers, the effect seen may have evened out or functional outcomes could have been different. In addition, the nature of the FHP device may have affected the ability to achieve an ideal reduction and cast fixation. However, as both groups used the same device, this limitation has been eliminated in the current study. “
Conclusion section, it should be softer.
Response: we thank the reviewer for the comment. Comment accepted and the conclusions section was edited.
Page 17 line 354 and below:
“Focused continues PEMF using a novel new device, the FHP, was safe to use in distal radius fracture treated with a cast. It demonstrated positive effect on fracture healing and positive short-term effects on functional outcomes. Furthermore, there were no complications related to the FHP device. Our results suggest that early addition of PEMF treatment may accelerate bone healing and, which could lead to a shorter cast immobilization, thus allowing an earlier return to daily life activities and work. Additionally, high compliance to the treatment is expected since the FHP device does not require any action by the patient. “

Round 2
Reviewer 2 Report
What is a bone that heals clinically?
Response: we thank the reviewer for this question. Several studies are cited in the text, one main study is a systematic review and meta‐analysis of randomized controlled trials, by Peng et al. they reported that PEMF treatment would increase overall healing rate with risk ratio of 1.22 (95% CI = 1.10–1.35). subgroup analysis was also conducted, showing increased healing rate in both long (femur, tibia, radius) and irregular bones. Additionally, in terms of pain, the pooled results of seven studies indicated that PEMF can alleviate pain.
Peng L, Fu C, Xiong F, Zhang Q, Liang Z, Chen L, He C, Wei Q. Effectiveness of Pulsed Electromagnetic Fields on Bone Healing: A Systematic Review and Meta-Analysis of Randomized Controlled Trials. Bioelectromagnetics. 2020 Jul;41(5):323-337. doi: 10.1002/bem.22271. Epub 2020 Jun 3. PMID: 32495506.
121 you write "accelerate healing both clinically and radiologically". You also use this term L18, L64, and L67. What do you mean by this? How can you assess clinical healing of the bone?
This sentence is also written twice
L121 you are writing “accelerate healing both clinically and radiologically”. You are also using this term L18, L64 and L67. What does that mean. How could you assess a clinical healing of the bone ?
Also this sentence is written two times “The 65 primary aim of the study was to determine whether the use of FHP by means of PEMF in 66 acute DRFs will accelerate healing both clinically and radiologically.”
-Why did you perform a numerical analysis on 20 patients to find a statistical difference?
Response: we thank the reviewer for this comment. Indeed, our study consists of a small group of patients, which we addressed in the Limitations section. Statistical analysis was performed by a statistician, and even with the small sample size, statistically significant differences were found.
This question was worded incorrectly. Did you do a sample size calculation before your study. This part is mandatory in RCT.
-Why did you measure grip strength but not report the results?
Response: we thank the reviewer for this comment. The results regarding grip strength are in the results section, under the subheading “Functional assessment.”
Page 12, Line 245 and below:
“Hand grip strength was measured after a cast removal. No statistically significant differences were found between the groups.”
Please add the value (mean and SD) so that these data can be reused in another study.
-What about the analysis of the comparability of the groups after the "lost to follow-up" in 13-14 patients?
Response: we thank the reviewer for pointing this important issue. The primary outcome was fracture union at 4-weeks based on CT scans. The data and statistical analysis were performed on patients in both group who completed the study protocol and were available for the final follow up.
My question was: if you remove 7 patients over 21 years of age, about 33% of the population, will the two populations still be comparable. Analyze your Table 1 with a statistical test and repeat the analysis it with the new population of 13 and 14 patients.
What is the main outcome? 4 weeks, 6 weeks? You cannot draw conclusions if you have a vague question.
Response: we thank the reviewer for this question. Comment accepted and addressed.
The primary outcome was defined, and text now reads:
Page 8, line 141 and below:
“Primary outcome was fracture union at 4-weeks based on CT scans.”
In the results section
What is group A or B?
Response: we thank the reviewer for this question. The information is detailed in the methods section. Page 6, Line 108:
“Eligible patients were randomly assigned to one of the two groups; Group 1 (Active): Standard treatment + Active FHP and Group 2 (Control): Standard treatment + Sham FHP.”
We have edited Table 2.
|
Group |
No union (0%-24%) |
Partial union (25%-74%) |
Union (75%-100%) |
|
Active |
0 |
5 |
9 |
|
Control |
1 |
8 |
3 |
Why was 1+8+3 = 12 patients given in the control group and in the flowchart it is 13 patients.
And why do you write in the flowchart under the active group "analyzed 14; excluded =3" 20-14 = 7 so you excluded 7 patients and not 3.
Please could you report CT image of no-union ; partial union and union
Table 2. Fractures categorization by extent of the inion at 4 weeks.
Did you mean Union not “inion”
Could you show the analysis between observers.
Response: we thank the reviewer for this valuable comment. Agreements between raters were tested by the paired t-test. To reduce the within variability in RUSS scale, we choose the mean and the median value from the 3 raters. Agreement between the two orthopedic surgeons was statistically significant, however, agreement between them and the radiologist was low. This data is shown in Figure 3.
This is not a proper agreement between the rater. You should use Cohen's kappa. Could you please do that.
Author Response
REPLIES TO THE REVIEWERS
Original Reviewers’ comments are copied here in italics.
We wish to thank the reviewer for the in‑depth analysis of our work and for raising several important points that needed clarification. We appreciate the time and effort expended on our behalf. We addressed each issue that was raised as follows:
Reviewer comments:
121 you write "accelerate healing both clinically and radiologically". You also use this term L18, L64, and L67. What do you mean by this? How can you assess clinical healing of the bone? This sentence is also written twice. L121 you are writing “accelerate healing both clinically and radiologically”. You are also using this term L18, L64 and L67. What does that mean. How could you assess a clinical healing of the bone? Also, this sentence is written two times “The 65 primary aim of the study was to determine whether the use of FHP by means of PEMF in 66 acute DRFs will accelerate healing both clinically and radiologically.”
Response: We thank the reviewer for this valuable comment. Our intention, from a clinical point of view, was to assess if there was pain at the fracture site and to evaluate range of motion and strength. We agree with the problem in the term and have changed the text accordingly.
Page 3, Line 30 and below:
“Background: This study aimed at determining whether the application of a novel new method of generating pulsed electromagnetic field (PEMF), the Fracture Healing Patch (FHP), accelerates healing of acute distal radius fractures (DRF) when compared to a sham treatment.”
Page 5, Line 78 and below:
“The primary aim of the study was to determine whether the application of the FHP generated PEMF as an adjuvant to immobilization for acute DRF, treated non-operatively, will accelerate bone healing. It was hypothesized that PEMF would accelerate extent of the fracture union by up to 30% as assessed by CT scans.”
Page 7, Line 131:
“Primary objectives of the study were to determine whether the use of FHP by means of PEMF in acute DRFs will accelerate bone healing.”
Did you do a sample size calculation before your study. This part is mandatory in RCT.
Response: we thank the reviewer for this question. Power analysis was performed before the study. We added this information to the statistical analysis section:
Page 10, Line 199 and below:
“Power analysis was conducted with an expected outcome difference of 30% in the extent of the fracture union assessed by CT at 4 weeks as compared to the control group. The alpha error level was set at 5% (two-sided significance level); power was set at 80%. Including an anticipated dropout rate of 10%, this resulted in a sample size of 23 patients per group. Data were analyzed with IBM SPSS statistics software version 28.0. (SPSS Inc. Headquarters, 233 S. Wacker Drive, 11th floor Chicago, Illinois 60606, USA). The significance levels were set at 0.05.”
Why did you measure grip strength but not report the results? Please add the value (mean and SD) so that these data can be reused in another study.
Response: we thank the reviewer for this comment. The results regarding grip strength are in the results section, under the subheading “Functional assessment.” Data regarding mean and SD were added.
Page 12, Line 247 and below:
“Hand grip strength was measured after a cast removal. At 6 weeks, the mean grip strength in the active group were 7.49 ±1.84 Kg vs 6.33±1.86 Kg in the control group. (p=0.684) At 12 weeks, the mean grip strength in the active group were 14.22 ±2.67 Kg vs 8.25±2.19 Kg in the control group. (p=0.114)”
What about the analysis of the comparability of the groups after the "lost to follow-up" in 13-14 patients?
My question was: if you remove 7 patients over 21 years of age, about 33% of the population, will the two populations still be comparable. Analyze your Table 1 with a statistical test and repeat the analysis it with the new population of 13 and 14 patients.
Response: We thank you for your question. We appreciate your attention to detail and your concern for the validity of our results. We chose to demonstrate the demographic data of patients in both groups as it provides important context and helps to ensure that the two groups were similar at baseline. By doing so, we aimed to minimize the potential impact of any baseline differences on the results of our study.
However, we only included patients who completed the study protocol and included in the final analysis because patients who dropped out or were lost to follow-up could have different characteristics and thus, their inclusion in the analysis could affect the validity of our results. We hope that this explanation addresses your concerns. As per your request, please see below the table which includes the population that completed FU and was analyzed:
|
Female |
Male |
Total Average of Age |
Total Count of Sex |
|||
|
Row Labels |
Average of Age |
Count of Sex |
Average of Age |
Count of Sex |
||
|
A |
59 |
8 |
58 |
5 |
59 |
13 |
|
B |
60 |
11 |
55 |
3 |
59 |
14 |
Why were 1+8+3 = 12 patients given in the control group and in the flowchart it is 13 patients. And why do you write in the flowchart under the active group "analyzed 14; excluded =3" 20-14 = 7 so you excluded 7 patients and not 3.
Response: We would like to express our gratitude to you for bringing the errors in Table 2 and Figure 1 to our attention. There was a typo in the table and in the flowchart. The table and figure have been corrected accordingly.
The figure please see the attachment.
|
Group |
No union (0%-24%) |
Partial union (25%-74%) |
Union (75%-100%) |
|
Control |
1 |
9 |
3 |
|
Active |
0 |
5 |
9 |
Please could you report CT image of no-union, partial union, and union.
Response: we thank the reviewer for this suggestion. We understand the importance of including images to support our findings. However, as mentioned in the method section (Page 8, line 163): “the evaluation of the extant of fracture union was performed in each of the axial cuts, then calculating the average. The extent of union was quantified as described by Singh et al [15]. Fractures were categorized as following: no union (0% to 24% of the continuity of the trabecular bridging across the whole width of the distal radius), partial union (25% to 74% trabecular bridging) or union (75% to 100% trabecular bridging).” Therefore, we think that a single image of a CT scan would not be representative for this purpose.
Did you mean Union not “inion”?
Response: we thank the reviewer for noticing the Typo. We meant union and we change the text accordingly.
Page 12, Line 237:
“Table 2. Fractures categorization by extent of the union at 4 weeks. “
This is not a proper agreement between the raters. You should use Cohen's kappa. Could you please do that.
Response: we thank the reviewer for this valuable comment. We used the Friedman test to assess inter-observer agreement among the three observers in this study.
The Friedman test is particularly useful in situations where the normality assumption of parametric tests cannot be met. (appropriate for comparing the rankings of multiple raters in the absence of a gold standard).
Page 10, Line 210:
“Agreements between raters were tested by the Friedman test. To reduce the within variability in RUSS scale, we choose the mean and the median value from the 3 raters”.
